# SAGCI-System: Towards **Sa**mple-Efficient, **G**eneralizable, **C**ompositional, and **I**ncremental Robot Learning

Jun Lv[*1], Qiaojun Yu[*1], Lin Shao[*2], Wenhai Liu[3], Wenqiang Xu[1] and Cewu Lu[1]

*Abstract*— Building general-purpose robots to perform a diverse range of tasks in a large variety of environments in the physical world at the human level is extremely challenging. According to [1], it requires the robot learning to be sample-efficient, generalizable, compositional, and incremental. In this work, we introduce a systematic learning framework called SAGCI-system towards achieving these above four requirements. Our system first takes the raw point clouds gathered by the camera mounted on the robot's wrist as the inputs and produces initial modeling of the surrounding environment represented as a file of Unified Robot Description Format (URDF). Our system adopts a learning-augmented differentiable simulation that loads the URDF. The robot then utilizes the interactive perception to interact with the environment to online verify and modify the URDF. Leveraging the differentiable simulation, we propose a model-based learning algorithm combining object-centric and robot-centric stages to efficiently produce policies to accomplish manipulation tasks. We apply our system to perform articulated object manipulation tasks, both in the simulation and the real world. Extensive experiments demonstrate the effectiveness of our proposed learning framework. Supplemental materials and videos are available on our project webpage https://sites.google.com/view/egci.

## I. INTRODUCTION

Building general-purpose robots to perform a diverse range of tasks in a large variety of environments in the physical world at the human level is extremely challenging. Consider a robot operating in a household. The robot faces various challenges. It would need to perform a broad range of tasks, such as preparing food in the kitchen and cleaning the floor in the living room. In the process, it must be able to handle the immense diversity of objects and variability of unstructured environments. Moreover, it would also need to quickly learn online to accomplish new tasks requested by humans. According to [1], building such an intelligent system requires the robot learning to be: sample-efficient, which means the robot needs to master skills using few training samples; generalizable, which requires the robot to accomplish tasks under unseen but similar environments or settings; compositional, which indicates that the various knowledge and skills could be decomposed and combined to solve exponentially more problems; and incremental, which requires new knowledge and abilities could be added over time. However, deep learning/deep reinforcement learning fails to meet these requirements [1].

We propose a robot learning framework called **SAGCI-System** aiming to satisfy the above four requirements. In our setting, the robot has an RGB-D camera mounted on its wrist and can perceive the surrounding environment. Our system first takes the raw point clouds through the camera as the inputs, and produces initial modeling of the surrounding environment represented as a file of URDF [2]. Based on the initial modeling, the robot

leverages the interactive perception (IP) [3] to interact with the environments to online modify the URDF. Our pipeline adopts a learning-augmented differentiable simulation that loads the URDF. We augment the differentiable simulation with differential neural networks to tackle the simulation error. We propose a novel model-based learning algorithm combining object-centric and robot-centric stages to efficiently produce policies to accomplish manipulation tasks. The policies learned through our pipeline would be sample-efficient and generalizable since we integrate the structured physics-informed "inductive bias" in the learning process. Moreover, the knowledge and robotic skills learned through our model-based approaches are associated with the generated URDF files which are organized with hierarchical structures, straightly resulting in compositionality and incrementality.

Our primary contributions are: (1)we propose a systematic learning framework towards achieving sample-efficient, generalizable, compositional, and incremental robot learning; (2)we propose neural network models to generate the URDF file of the environment and leverage the interactive perception to iteratively make the environment model more accurate; (3) we introduce a model-based learning approach based on the learning-augmented differentiable simulation to reduce the sim-to-real gap; (4) we conduct extensive quantitative experiments to demonstrate the effectiveness of our proposed approach; (5) we apply our framework on real-world experiments to accomplish a diverse set of tasks.

## II. RELATED WORK

We review literature related to the key components in our approach including the interactive perception, differential simulation, and model-based reinforcement learning/control, and describe how we are different from previous works.

### A. Interactive Perception

Interactive perception (IP) is related to an extensive body of work in robotics and computer vision. It has enjoyed success in a wide range of applications, including object segmentation [4, 5], object recognition [6], object sorting [7, 8], and object search [9, 10]. For a broader review of IP, we refer to [3]. Hausman et al. [11] introduced a particle filter-based approach to represent the uncertainty over articulation models and selected actions to efficiently reduce the uncertainty over these models. Martin and Brock [12] presented an IP algorithm to perform articulated object estimation on the fly by formulating the perception problem as three interconnected recursive estimation filters. In this work, we leverage IP to construct, verify and modify the modeling of the surrounding environments.

### B. Differential Simulation

Differentiable simulation provides gradients in physical systems for learning, control, and inverse problems. Leveraging recent advances in automatic differentiation methods [13–18], a number of differentiable physics engines have been proposed to solve system identification and control problems [15, 19–21, 21, 22, 22–28]. An alternative approach towards a differentiable physics engine is to

*The authors contributed equally

[1]Jun Lv, Qiaojun Yu, Wenqiang Xu, Cewu Lu are with Department of Computer Science, Shanghai Jiao Tong University, China. {lyujune_sjtu,yqjllxs,vinjohn,lucewu}@sjtu.edu.cn

[2] Lin Shao is with Artificial Intelligence Lab, Stanford University, USA. lins2@stanford.edu

[3]Wenhai Liu is with School of Mechanical Engineering, Shanghai Jiao Tong University, China. sjtu-wenhai@sjtu.edu.cn

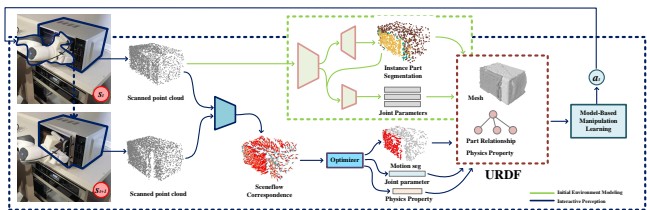

**Fig. 1:** The proposed pipeline takes the raw point clouds captured by the RGB-D camera mounted on the robot as the inputs, and produces initial modeling of the surrounding environment represented as a URDF. Then robot leverages the interactive perception to online verify and modify the URDF. We also propose a novel model-based manipulation learning method to accomplish manipulation tasks.

approximate physics with neural networks leveraging data-driven approaches [29–31]. These neural networks are implicitly differentiable, but the learned networks might not satisfy the physical dynamics. The simulation quality may also degenerate beyond the training data distribution. One line of works [32, 33] addresses these issues by augmenting the simulation with neural networks. In this work, we adopt the Nimble simulation [34] and augment the differentiable simulation with neural networks to address the sim-to-real gap while taking advantage of the differentiation to develop manipulation policies.

### C. Model-Based Reinforcement Learning

Model-based reinforcement learning (MBRL) is recognized with the potential to be significantly more sample efficient than model-free RL [35]. However, developing an accurate model of the environment is a challenging problem. Modeling errors degenerate the performances by misleading the policies to exploit the models' deficiencies. For a broader review of the field on MBRL, we refer to [36]. In this work, we develop a system to model the environment and integrate a learning-augmented differentiable simulation to reduce the modeling error. To mitigate the reward/cost function engineering, we propose a two-level (object-centric and robot-centric) procedures to generate manipulation policies.

### III. TECHNICAL APPROACH

Once deployed in an unstructured environment, the robot would begin to perceive the surrounding environment through an RGB-D camera mounted on its wrist. Our pipeline takes the raw point clouds as inputs, and produces initial modeling of the surrounding environment represented URDF. Based on the initial modeling, our robot leverages the interactive perception to online modify the URDF. Our pipeline adopts a learning-augmented differentiable simulation to tackle the modeling error. Leveraging the differentiable simulation, we propose a model-based learning algorithm combining object-centric and robot-centric stages to efficiently produce policies to accomplish manipulation tasks. An overview of our proposed method is shown in Fig. 1. We describe the above modules in the following subsections.

### A. Environment Initial Modeling

With an RGB-D camera mounted at the wrist, the robot receives point clouds at time step $t$ as $\{\mathcal{P}_t^i\}_{i=1}^N \in \mathcal{R}^{N \times 3}$, where the $N$ is the total number of points. In this subsection, we would discuss our pipeline, which takes $\{\mathcal{P}_t^i\}_{i=1}^N$ as the inputs to initially model the surrounding environment.

We use the file of URDF to represent the environment, which could be directly loaded into most popular robotic simulations such as Mujoco [37], Bullet [38], and Nimble [34] for model-based control/learning/planning. In this work, we only care about one

object at a time. The generated URDF file should only contain descriptions of a single object's links and joints. Each link is a rigid body part containing the mesh files describing the geometry shape and physical attributes such as the mass value, the inertial matrix, and the contact friction coefficients. Each joint describes the relationship and its parameter between two links. For more explanations about the URDF, we refer to [39].

We first describe how we generate these descriptions of the object's links. We develop a part-level instance segmentation model that takes the point clouds $\{\mathcal{P}_t^i\}_{i=1}^N$ as input, and outputs a part-level instance segmentation mask denoted as $\{\mathcal{M}_t^i\}_{i=1}^N$ where $\mathcal{M}_t^i \in \{1, 2, ..., K\}$. $K$ is the number of parts. We then segment the raw point cloud $\{\mathcal{P}_t^i\}_{i=1}^N$ into a set of groups denoted as $\{\mathcal{G}_t^j\}_{j=1}^K$. Here each group corresponds to a link in the URDF. Then we would generate a watertight mesh based on the group through ManifoldPlus [40]. Based on the raw input point clouds of each group, our model also estimates the physical proprieties of the link. Note that these values may not be correct, and we would modify these properties at the IP stage in Sec. III-B.

Then we would explain how to produce the corresponding joints in the URDF. First, given $K$ links, we would develop a joint relationship model that takes the segmented point clouds of the link $u$ and the link $v$, where $u, v \in \{1, 2, ..., K\}$, to estimate their pairwise joint relationship denoted as $\mathcal{J} \in \mathcal{R}^{K \times K \times 4}$. The predicted result item $\mathcal{J}(u, v)$ contains the probability for the "None", "Revolute", "Prismatic", "Fixed" joint between two links. Here "None" indicates that two links do not contain a direct joint relationship. The model also predicts the joint spatial descriptions denoted as $\mathcal{C} \in \mathcal{R}^{K \times K \times 9}$, contains joint axis, origin, and orientation for each pair. Till now, we get a complete directed graph through $K$ links. To compose a URDF, we adopt a greedy algorithm to find a directed tree denoted as $\mathcal{E} = \{u_i, v_i\}_{i=1}^{K-1}$ from the complete graph, which contains $K - 1$ joints. Up to now, we have generated an initial URDF file to describe the environment. Due to the page limit, we report the detailed pipeline in the supplementary material.

### B. Interactive Perception

Although we have estimated a URDF file based on the raw point clouds, there are always modeling errors during the above-mentioned method, due to imperfect recognition. We propose a pipeline to train the robot to take advantage of the Interactive Perception (IP) to modify the URDF. We first discuss what modeling parameters are re-estimated through the IP and then introduce the pipeline of how to update these modeling parameters.

*a) Model Parameter:* We would update the following model parameters: (1) the joint type $\mathcal{J}$; (2) the joint spatial descriptions $\mathcal{C}$; (3) each link's mask segmentation $\mathcal{M}$ and the corresponding mesh file; (4) the physical attributes $\alpha^{sim}$. We denote the model parameter set as $\mathcal{Z} = \{\mathcal{J}, \mathcal{C}, \mathcal{M}, \alpha^{sim}\}$, and we also denote $\mathcal{E}$ to describe the URDF tree structure.

Our system receives the raw point cloud $\{\mathcal{P}_t^i\}_{i=1}^N$ at the time step $t$, and generates an action denoted as $a_t^{IP}$. After the action $a_t^{IP}$ are executed, we observe the state difference between the simulation and the real world. Minimizing their difference would lead to a more accurate $\mathcal{Z}$. To accomplish this, we need to establish the correspondence between the real world and the simulation at first.

*b) Correspondence in Simulation:* In the simulation, we could directly calculate the 3D positions of $\{\mathcal{P}_t^i\}_{i=1}^N$ at time $t + 1$, denote as $\{\overline{\mathcal{P}}_{t+1}^i\}_{i=1}^N$, via a forward function $\mathcal{F}^{Sim}$, given model parameters $\mathcal{Z}, \mathcal{E}$ and action $a_t$.

$$\overline{\mathcal{P}}_{t+1}^i = \mathcal{F}^{Sim}(\mathcal{P}_t^i, \mathcal{Z}, \mathcal{E}, a_t^{IP}) \tag{1}$$

As shown in Eqn 1, each point $\mathcal{P}_t^i$ is associated with $\overline{\mathcal{P}}_{t+1}^i$ through the forward simulation.

*c) Correspondence in Real World:* Through the RGB-D camera, the robot in the real world receives a new point cloud at the time $t+1$ denoted as $\{\mathcal{P}_{t+1}^i\}_{i=1}^N$. We train a scene flow [41] model that takes raw point clouds $\{\mathcal{P}_t^i\}_{i=1}^N$ and $\{\mathcal{P}_{t+1}^i\}_{i=1}^N$ as inputs, and outputs the scene flow $\{\mathcal{U}_t^i\}_{i=1}^N$. We then calculate the 3D positions of the point clouds $\{\mathcal{P}_t^i\}_{i=1}^N$ at the time $t+1$, denoted as $\{\mathcal{P}_t^i + \mathcal{U}_t^i\}_{i=1}^N$. Each point $\mathcal{P}_t^i + \mathcal{U}_t^i$ searches for the nearest point in $\{\mathcal{P}_{t+1}^i\}_{i=1}^N$. We denote these found points in $\{\mathcal{P}_{t+1}^i\}_{i=1}^N$ as $\{\widetilde{\mathcal{P}}_{t+1}^i\}_{i=1}^N$. $\{\widetilde{\mathcal{P}}_{t+1}^i\}_{i=1}^N$ is point-wise correspondence to $\{\mathcal{P}_t^i\}_{i=1}^N$. We denote the real-world forward function $\mathcal{F}^{real}$ as,

$$\widetilde{\mathcal{P}}_{t+1}^i = \mathcal{F}^{real}(\mathcal{P}_t^i, \mathcal{U}_t^i, \{\mathcal{P}_{t+1}^i\}_{i=1}^N) \tag{2}$$

*d) Model Parameter Optimization:* Given $\mathcal{P}_t^i$, we can compute its 3D position at next time step in both simulation and real world by $\mathcal{F}^{sim}$ and $\mathcal{F}^{real}$, which is $\overline{\mathcal{P}}_{t+1}^i$ and $\widetilde{\mathcal{P}}_{t+1}^i$ respectively. Accurate model parameters lead to a small difference of them. We denote the distance between $\{\overline{\mathcal{P}}_{t+1}^i\}_{i=1}^N$ and $\{\widetilde{\mathcal{P}}_{t+1}^i\}_{i=1}^N$ as the $\mathcal{L}_{t+1}$

$$\mathcal{L}_{t+1}(a_t^{IP}, \mathcal{Z}, \mathcal{E}) = \frac{1}{N}\sum_{i=1}^N \|\overline{\mathcal{P}}_{t+1}^i - \widetilde{\mathcal{P}}_{t+1}^i\|^2 \tag{3}$$

The $\mathcal{L}_{t+1}$ is adopted to measure the modeling quality. Since the function $\mathcal{F}^{sim}$ is differentiable with respect to the model parameters $\mathcal{Z}$. We could optimize the $\mathcal{Z}$ through gradient descent and find new model parameter set $\mathcal{Z}' = \{\mathcal{J}', \mathcal{C}', \mathcal{M}', \alpha^{sim\prime}\}$ and obtain $\mathcal{E}'$ from $\mathcal{E}$ with the help of newly optimized $\mathcal{Z}'$. In this way, we gradually approach accurate modeling of the environment.

$$\mathcal{Z}' = \mathcal{Z} - \lambda \frac{\partial \mathcal{L}_{t+1}}{\partial \mathcal{Z}} \tag{4}$$

*e) Policy Network:* We introduce a deep reinforcement learning network which tasks as inputs the raw point cloud $\{\mathcal{P}_t^i\}_{i=1}^N$ and model parameter set $\mathcal{Z}$, and outputs an action denoted as $a_t^{IP}$. Here the $a_t^{IP}$ contains a discrete action which is the link id on the predicted URDF, and a continuous action that determines the goal state change of the relative joint in the simulation world. How to generate the associated robot manipulation actions will be discussed in III-D. We define the modeling quality improvement in Eqn. 5 as the reward of taking the action $a_t^{IP}$ given the current state.

$$r_t = \mathcal{L}_{t+1}(a_t^{IP}, \mathcal{Z}, \mathcal{E}) - \mathcal{L}_{t+1}(a_t^{IP}, \mathcal{Z}', \mathcal{E}') \tag{5}$$

Leveraging the IP, our pipeline can verify and modify the URDF file to achieve better environment modeling. Detailed explanation of each part is put in the supplementary material.

### C. Sim-Real Gap Reduction Through augmenting differential simulation with neural networks

Even if the analytical model parameters have been provided, rigid-body dynamics alone often does not exactly predict the motion of mechanisms in the real world [33]. In the interactive perception stage, our pipeline would optimize the modeling parameters to reduce the differences between the simulation and the real world. However there are always discrepancy remaining. To address this, we propose a simulation that leverages differentiable physics models and neural networks to allow the efficient reduction of the sim-real gap. We denote $s_t^{sim}$ and $s_t$ as current state of simulation and real world respectively. We develop a neural network model denoted *NeuralNet* to predict the residual change of the next state based on the current state, the current action, and the calculated next state from the Nimble simulation. In the real world, our robot would take action $a_t$ based on the state $s_t$ and gather the transition

tuple $(s_t, a_t, s_{t+1})$. Meanwhile in the simulation, our robot would also take the same action $a_t$ based on the state $s_t$ and gather the transition tuple $(s_t^{sim}, a_t, s_{t+1}^{sim})$. Here $s_t^{sim}$ and $s_t$ are the same. We define a loss denoted as $\mathcal{L}^{aug}$ to measure the difference between $s_{t+1}^{sim}$ and $s_{t+1}$. Due the page limit, we report the detailed process to calculate the $\mathcal{L}^{aug}$ in the supplementary material.

### D. Model-Based Manipulation Learning

In this subsection, we discuss how our system produces robotic manipulation actions to accomplish a given task denoted as $\mathcal{T}$. For example, the robot may receive a task request from Sec. III-B to open the microwave by $\theta$ degree, which is the action $a^{IP}$ defined in Sec. III-B during the interactive perception stage. We first train the robot to accomplish the task $\mathcal{T}$ in the differential simulation. We then record these robotic manipulation sequences from the simulation denoted as $\{(s_t^{sim}, a_t^{sim})\}_{t=0}^{T-1}$, and utilize these manipulation sequences to guide the robotic execution in the real-world.

*a) Manipulation in the Simulation:* Based on the modeling of the environment, we propose two-level procedures to guide the robot in the simulation to reach the target goal $\mathcal{G}(\mathcal{T})$ which indicates the success of the task $\mathcal{T}$.

**Object-centric setting**. In this setting, the robot is not loaded into the simulation and we would directly control the objects to find a feasible path to reach the goal $\mathcal{G}(\mathcal{T})$. We denote the state and the action in this object-centric setup to be $x_t$ and $u_t$. Here $x_t = [q_t, \dot{q}_t]$ which contains the object current joint value $q_t$ and the joint velocity $\dot{q}_t$. $u_t$ represents the external forces exerted directly to control the objects.

We formulate the problem of finding a feasible path to reach the goal as an optimal control problem with the dynamic function denoted as $\mathcal{F}^{sim}$ and the cost function $l^o$ shown as below. Here $T$ is the maximal time step.

$$\mathcal{L}_t^o(\mathcal{T}) = \sum_{t=0}^{T-1} l_t^o(x_t, u_t; \mathcal{T}) \tag{6}$$

$$s.t. \quad x_{t+1} = \mathcal{F}^{sim}(x_t, u_t), x_0 = x_{init} \quad . \tag{7}$$

We could find a solution to the optimization problem leveraging the differentiable simulation. Detailed explanations are reported in the supplementary material. We record the corresponding procedure sequences denoted as $\{x_t^*, u_t^*\}_{t=0}^{T-1}$. The sequences reflect how the objects should be transformed in order to successfully reach the goal $\mathcal{G}(\mathcal{T})$, which are used to guide robot actions at the next level.

**Robot-centric setting**. After gathering the $\{x_t^*, u_t^*\}_{t=0}^{T-1}$ in the object-centric setting, we load the robot into the simulation and start the robot-centric procedure, in which we find the robotic action sequences to accomplish the task $\mathcal{T}$. Note that in the robot-centric setting, the state $s_t^{sim} = [x_t; q_t^r, \dot{q}_t^r]$ is composed of two components: the objects' state denote as $x_t$, which are the joint position $q_t^o$ and joint velocity $\dot{q}_t^o$; the robot's joint position $q_t^r$ and joint velocity $\dot{q}_t^r$. The action $a_t^{sim}$ in the robot-centric setting is the robot control forces denoted as $\tau$. Note that $a_t^{sim}$ does not contain $u_t$ used in the object-centric setup meaning that we does not directly control object in the robot-centric setting.

$$\mathcal{L}_t^r(\mathcal{T}) = \sum_{t=0}^{T-1} l_t^r(s_t^{sim}, a_t^{sim}; \{x_t^*\}_{t=0}^{T-1}, \{u_t^*\}_{t=0}^{T-1}, \mathcal{T}) \tag{8}$$

$$s.t. \quad s_{t+1}^{sim} = \mathcal{F}^{sim}(s_t^{sim}, a_t^{sim}), s_0^{sim} = s_{init} \\ a_t^{sim} = \pi_\theta(s_t^{sim}) \tag{9}$$

We formulate the problem of directly find the sequence of robotic actions $\{a_t^{sim}\}_{t=0}^{T-1}$ to accomplish the task $\mathcal{T}$ as an optimization

problem. The cost function is denoted as $l^r$ and the policy parametered by $\theta$ is $\pi_\theta$. Due to the page limit, we put the detailed explanation of how to optimize $\mathcal{L}^r$ and $\pi_\theta$ in the supplementary material. We denote the corresponding sequences as $\{s_t^*, a_t^*\}_{t=0}^T$. Note that we could switch back to the Object-centric setting if some hard constraints are met such as the robot reach its joint limit or is near to a potential collision.

*b) Guided Manipulation in the Real world:* After receiving the sequence of states $\{s_t^{sim*}, a_t^{sim*}\}_{t=0}^T$ and the policy $\pi_\theta$, we would execute the learned policy $\pi_\theta(s^t)$ and the state will change to $s_{t+1}$. Detailed explanations are in supplementary material. Meanwhile, we record the transition tuple in the real world $(s_t^r, a_t^r, s_{t+1}^r)$. If the current episode fails to accomplish the task $\mathcal{T}$ in the real world. These states would be added into the memory buffer to improve the quality of the model as described in Sec. III-C.

## IV. EXPERIMENTS

In this work, we develop a learning framework aiming to achieve sample-efficient, generalizable, compositional, and incremental robot learning. Our experiments focus on evaluating the following question: (1) How effective is our proposed interactive perception framework? (2) Whether our approach is more sample-efficient and has better generalization compared to other approaches? (3) How useful is our model in zero/few-shot learning, task composition/combination, long-horizon manipulations tasks?

### A. Experimental Setup

*a) Simulation:* We use PyBullet [38] to simulate the real world which is different from our differentiable simulation based on Nimble [34]. We conduct our experiments using six object categories from the SAPIEN dataset [42], which are box, door, microwave, oven, refrigerator, and storage furniture. We select 176 models in total, 143 models for training, and 33 models for testing.

*b) Real World:* We also set up the real world experiment. We mount an RGB-D camera RealSense on the wrist of the 7-Dof Franka robot. Due to the page limit, we put the real-world experiment in the supplementary material.

### B. Evaluating the Performance of Interactive Perception

In this subsection, we compare the modeling qualities before and after the interactive perception operations. Given the initial modeling of the environment, we verify and modify each part on the predicted URDF with a sequence of interactions, and online optimize the URDF by each interaction step. To evaluate the performance, We compute the average precision under IoU 0.75 of instance part segmentation (AP75), joint type classification accuracy (Acc.), joint orientation error (Rot.), and joint translation error (Tran.). Comparing the results of the "init" and "opt" in Tab. I, the pipeline can significantly improve the URDF quality after interactive perception.

**TABLE I:** Performance of interactive perception.

| | AP75 ↑ | | Acc. ↑ | | Rot. ↓ | | Tran. ↓ | |
| | init | opt | init | opt | init | opt | init | opt |
|---|---|---|---|---|---|---|---|---|
| Door | **0.748** | 0.714 | 0.975 | **0.975** | 13.713 | **2.252** | 15.806 | **3.670** |
| Microwave | 0.800 | **0.939** | 1.000 | **1.000** | 11.562 | **4.230** | 15.439 | **3.788** |
| Box | **0.895** | 0.822 | 0.877 | **0.907** | 10.143 | **4.530** | 15.241 | **7.899** |
| Oven | 0.773 | **0.875** | 1.000 | **1.000** | 23.525 | **6.589** | 18.731 | **10.066** |
| Fridge | 0.584 | **0.814** | 0.989 | 0.955 | 12.174 | **5.331** | 18.000 | **7.820** |
| Storage | 0.499 | **0.519** | 0.924 | **0.949** | 11.564 | **9.831** | 33.510 | **7.404** |
| Overall | 0.717 | **0.781** | 0.961 | **0.964** | 13.780 | **5.461** | 19.455 | **6.775** |

### C. Evaluating the Sample-Efficiency and Generalization

We compare our proposed approach with two popular model-free RL algorithms: SAC [43] and TD3 [44] on articulated object manipulation tasks. The tasks are to teach the robot to open the articulated objects of six classes. We post the result of opening the microwaves and more results on other categories are put in the supplementary material.

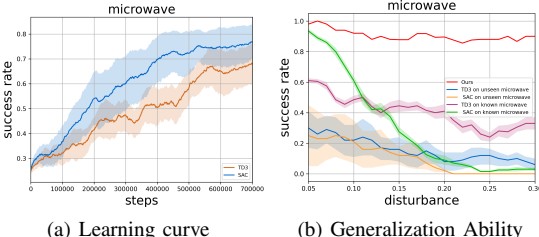

(a) Learning curve          (b) Generalization Ability

**Fig. 2:** (a) Learning curves of SAC and TD3. (b) Comparison of SAC, TD3 and our method in terms of the generalization on the microwave

For the model-free RL approaches, we describe the definitions of the states, actions, and the reward functions in the supplementary material. Fig. 2(a) shows the average success rate of the SAC and TD3 in the training stage. After training 700k steps, the average success rate of SAC models and TD3 models to open microwaves are about 80% and 70%, respectively. However, our pipeline achieves a success rate of around 90% after five interactive perception operations and training experiences/samples. Moreover, the modeling constructed by our approach could be adopted for other related tasks immediately. To evaluate the generalization abilities of different approaches, we change the articulated object's 6D pose by a 6D offset/disturbance. As shown in Fig. 2(b), with the increasing disturbance to the robot end-effector's initial 6D pose, the performance of SAC and TD3 decreases rapidly, especially in manipulating unseen microwaves. The results show that our method performs comparably to the baseline methods on the small disturbance and outperforms them on the large disturbance and on unseen microwaves, indicating a significantly better generalization ability.

### D. Evaluating the compositionality and incrementality

We put the experiments of this subsection in the supplementary material due to the page limit.

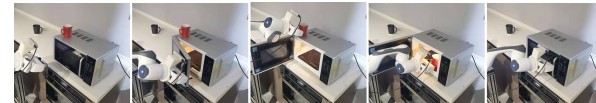

**Fig. 3:** Open the microwave, put the mug into it, and close it

## V. CONCLUSION

We present a learning system called **SAGCI-System** aiming to achieve sample-efficient, generalizable, compositional, and incremental robot learning. Our system first estimates an initial URDF to model the surrounding environment. The URDF would be loaded into a learning-augmented differential simulation. We leverage the interactive perception to online correct the URDF. Based on the modeling, we propose a new model-based learning approach to generate policies to accomplish various tasks. We apply the system to articulated object manipulation tasks. Extensive experiments in the simulation and the real world demonstrate the effectiveness of our proposed learning framework.

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
