# OpenReview forum: "SAGCI-System: Towards Sample-Efficient, Generalizable, Compositional, and Incremental Robot Learning"
_ICRA.org/2022/Workshop/Contact-Rich — ICRA 2022 Workshop: RL for Manipulation Oral_

### Official Review · Reviewer_E4RV · 2022-05-06
**Very interesting systems paper based on interactive perception for incremental learning, difficult to present in a short format.**

**Rating:** 7
**Confidence:** 3

**Review:**

This paper presents a framework aiming to accommodate incremental robot learning in diverse, human-centric environment where the variety of objects and tasks pose a major challenge. The authors are suggesting a system that starting from point clouds, composes descriptions of the surrounding objects and the joints that comprise them. Then, based on interactive perception, the initial estimates are refined to eventually be used in a differentiable simulation which can be utilized to learn manipulation policies in a sample-efficient manner.
Overall this is a really interesting paper that proposes a framework that could be very helpful in a lot of scenarios. The short format of the paper is not beneficial as a lot of details are skipped, making it difficult to fully realize the potential.
My (minor) concerns/ regarding the paper are:
1) Given the plethora of URDF files that will probably be generated in an average environment, how scalable is the computational power needed to be able to accommodate them?
2) On average, how long does an interactive perception episode last until the object parameters are identified? Is is realistic to deploy a model in rooms with multiple objects?
3) In reinforcement learning really necessary for these tasks? Why aren't more traditional techniques sufficient given a well-estimated model of the objects?

---

### Official Review · Reviewer_qjyS · 2022-05-09
**Review for Paper "SAGCI-System: Towards Sample-Efficient, Generalizable, Compositional, and Incremental Robot Learning"**

**Rating:** 9
**Confidence:** 4

**Review:**

**Summary**: This paper proposes a general framework for robot learning. The system first builds a "mental model" of the environment and then learns manipulation policy in this mental model. To build the model of the environment, the system first generates a URDF file representing the environment. The URDF file is loaded into a differentiable simulator and then improved by interactive perception. The differentiable simulator is augmented with a neural network to improve the accuracy of forward dynamics prediction. A novel model-based policy learning method is proposed to learn manipulation policy. The effectiveness of the framework is demonstrated on both simulated and real-world manipulation tasks.

Strengths and Weaknesses

- The proposed system consists of multiple modules to solve non-trivial sub-problems. The fact that the system works on the real robot is impressive.
- The proposed model-based method for learning manipulation policies utilizes the strengths of a differentiable simulator.

Comments

- I think it's interesting to see how the inaccuracy in the estimated urdf model affects the performance of manipulation policy. One way to see this is through a sim-to-sim experiment, in which a policy is trained based on the estimated urdf model and tested on the environment corresponding to the ground truth urdf.
- Can the system add new objects to the urdf file after the initial step? For example, in the "take the tap out of the drawer" task, the tap is occluded initially if the drawer is closed and can only be detected when the robot open the drawer.